# Contamination Status and Acute Dietary Exposure Assessment of Paralytic Shellfish Toxins in Shellfish in the Dalian Area of the Yellow-Bohai Sea, China

**DOI:** 10.3390/foods13030361

**Published:** 2024-01-23

**Authors:** Pei Cao, Lei Zhang, Yaling Huang, Shuwen Li, Xiaodan Wang, Feng Pan, Xiaojin Yu, Jinfang Sun, Jiang Liang, Pingping Zhou, Xiaomin Xu

**Affiliations:** 1China National Center for Food Safety Risk Assessment, Beijing 100022, China; caopei@cfsa.net.cn (P.C.); zhanglei@cfsa.net.cn (L.Z.); 15959575021@163.com (Y.H.); wangxiaodan@cfsa.net.cn (X.W.); panfeng@cfsa.net.cn (F.P.); liangjiang@cfsa.net.cn (J.L.); 2College of Food Science and Technology, HuNan Agricultural University, Changsha 410125, China; 3Department of Epidemiology and Biostatistics, School of Public Health, Southeast University, Nanjing 210009, China; lsw04047777@163.com (S.L.); xiaojinyu@seu.edu.cn (X.Y.); jinfangsun@seu.edu.cn (J.S.); 4Zhe Jiang Provincial Center for Disease Control and Prevention, Hangzhou 310051, China

**Keywords:** paralytic shellfish toxins, acute exposure assessment, shellfish, probabilistic risk assessment, seasonal variation, human health

## Abstract

The Yellow-Bohai Sea is an important semi-enclosed continental shelf marginal seas with an intensive aquaculture industry in China. The current study analyzed the contamination status and the time variations of paralytic shellfish toxins (PSTs) in shellfish between 2019 and 2020 from the Yellow-Bohai Sea in the Dalian area and estimated the acute health risks to consumers in China. A total of 199 shellfish samples (including 34 Pacific oysters, 25 Mediterranean blue mussels, 34 Manila clams, 36 bay scallops, 34 veined rapa whelks and 36 bloody clams) were analyzed from four representative aquaculture zones around the Yellow-Bohai Sea in Dalian. Among the samples, scallops and blood clams were the shellfish species with the highest detection rate of PSTs (94.4%), and the highest level of PSTs was detected in scallops with 3953.5 μg STX.2HCl eq./kg (μg STX.2HCL equivalents per kg shellfish tissue), followed by blood clams with 993.4 μg STX.2HCl eq./kg. The contents of PSTs in shellfish showed a time variation trend, and autumn was the season of concern for PST contamination in Dalian. For general Chinese consumers, the probability of acute health risks to shellfish consumers from dietary exposure to PSTs was around 13%. For typical consumers in coastal areas of China, especially those with higher shellfish intake, there was an acute health risk associated with exposure to PSTs through shellfish consumption during the occurrence of harmful algal blooms. It is suggested that the government continue to strengthen the monitoring of the source of PSTs and the monitoring of harmful algal blooms and give reasonable advice on shellfish consumption for consumers in coastal areas, such as not eating scallop viscera.

## 1. Introduction

With the increasing use of fertilizers in agriculture, discharge of industrial and domestic wastewater, coastal eutrophication, climatic change, the world trade market, overseas transportation and harmful algal blooms have become widespread marine environmental problems, particularly in large river estuaries and semi-enclosed bays [1,2]. Marine algal toxins are secondary metabolites produced by toxic marine algae that can accumulate in bivalve mollusks through filter-feeding behavior [3]. Paralytic shellfish toxins (PSTs) are a group of neurotoxic algal toxins that are produced mainly by marine dinoflagellates belonging to the genera *Alexandrium*, *Gymnodinium* and *Pyrodinium,* which are distributed throughout the world [4,5]. Depending on their chemical structure (different R_4_ groups), PSTs can be reorganized into several subgroups (Figure 1) [6]. For example carbamate (saxitoxin (STX), neoSTX, gonyautoxins (GTX1, GTX4, GTX2, GTX3)), N-sulfocarbamoyl (GTX5) and decarbamoyl toxins (dcSTX, dcneoSTX) [7,8]. As PSTs readily accumulate in filter-feeding bivalve mollusks, the potential risk of shellfish to consumers is considered [9,10].

PST poisoning events in Europe and Japan have been reported since the late 1970s and early 1980s and have mainly occurred in countries near the coastlines of the Atlantic and Pacific Oceans [11,12,13]. PST poisonings in some coastal cities have also been recorded in China in the past two decades [13,14]; for example, in 2017, 164 human intoxications were recorded in Zhangzhou City, Fujian Province, due to the presence of PSTs in shellfish (oysters and mussels) [13]. The main clinical manifestations of poisoning were muscle paralysis, difficulty breathing and weakness, and the average latency was 3 h; 8 out of the 10 shellfish samples collected had PSTs [13].

Additionally, some studies have shown that the PST levels in shellfish differ among different sea areas in China and that the PST levels are easily affected by red tides [15,16]. Because the Bohai Sea is a semi-enclosed inland sea, red tides occur more frequently. Du et al. collected 21 scallop samples from Dalian, Liaoning Province, and PSTs were detected in 19 samples with the highest concentration being 23.84 MU/g, which exceeded the limit of China’s national food safety standard (GB2733-2015) [15,17]. This study suggested that the presence of high PST levels in shellfish may be related to toxic algae [15]. However, national monitoring of PSTs in shellfish has been carried out in China since 2016, but all shellfish samples were collected from the local market, and data on PST contents during farming and fishing are lacking.

In 2009, 2004 and 2016, the European Food Safety Authority (EFSA) and the Joint FAO/WHO Expert Committee on Food Additives (JECFA) estimated the acute dietary exposure to PSTs in shellfish, and the results showed that consumers with high levels of shellfish consumption were concerned [18,19,20]. Most studies on acute dietary exposure to PSTs have adopted the method of point assessment, but the high percentile values of PST content in shellfish and shellfish consumption have been found in point assessments; for example, P99 of the PST content and P99 of shellfish consumption significantly overestimate these results [9,21]. Zhou et al. used the point assessment method to estimate the acute dietary exposure to PSTs in scallop consumers; this value exceeded the acute reference dose (ARfD) of 0.5 μg STX.2HCl eq./kg bw established by the EFSA [9]. However, studies on the accurate estimation of acute dietary exposure to PSTs among Chinese consumers, especially those who prefer to eat shellfish in coastal areas, are lacking.

Boundy M. and Turner A.D. et al. developed the HILIC liquid chromatography coupled to mass spectrometry in tandem (HILIC–LC–MS/MS) method for the analysis of PSTs [6,7]. This method was also used to analyze the PST content in shellfish in this study. The purpose of this study was to understand the contamination status and temporal variation in the contaminant levels of PSTs in shellfish in the Dalian area from the Yellow-Bohai Sea from 2019 to 2020. Moreover, for a more accurate assessment, a probabilistic approach was used to assess the health risks of acute dietary exposure to PSTs in shellfish for general consumers and consumers in coastal areas in China.

## 2. Materials and Methods

### 2.1. Chemicals and Reagents

The certified reference materials of GTX1&4, GTX2&3, dcGTX2&3, neoSTX, STX, dcSTX and GTX5 were purchased from the National Research Council-Institute for Marine Biosciences (Halifax, NS, Canada). HPLC-grade acetonitrile was obtained from TEDIA (Fairfield, OH, USA). HPLC-grade formic acid, ammonium formate and analytical grade ammonium hydroxide (>25%, *w*/*w*) were obtained from Thermo Fisher Scientific (Shanghai, China). The water used in this study was 18.2 MΩ cm^−1^ ultrapure water which was prepared by Millipore (Bedford, MA, USA).

### 2.2. Standard Solution Preparation

Mixed standard intermediate solutions of STX, neoSTX, dcSTX, GTX5, GTX1, GTX4, GTX2, GTX3, dcGTX2 and dcGTX3 were prepared with different concentrations of matrix solution (Table 1). All the standard solutions were stored at −20 °C and kept away from light.

### 2.3. Sample Collection

A total of 199 shellfish samples were collected, including 34 Pacific oysters *(Crassostrea gigas)*, 25 Mediterranean blue mussels *(Mytilus galloprovincialis)*, 34 Manila clams *(Ruditapes philippinarum), 36* bay scallops (*Argopecten irradians*), 34 veined rapa whelks *(Rapana venosa)* and 36 blood clams *(Scapharca subcrenata).* All these samples were collected from four representative aquaculture zones around the Yellow-Bohai Sea in the Dalian area from October 2019 to October 2020. The sampling sites were Qidingshan Street (Manila clams), Dalijia Street (bay scallops and Pacific oysters), Daweijia Street (Mediterranean blue mussels) and Xingshu Street (veined rapa whelks and blood clams), Jinzhou District, Dalian. In this study, according to the occurrence time of toxic microalgae in the Yellow-Bohai Sea, shellfish samples were collected once every two weeks from October 2019 to March 2020 and once a week from April 2020 to October 2020 (no sampling occurred in February and August due to COVID-19), for a total number of sampling times of 35.

After each sample was collected, the sediment on the shell of the shellfish samples was washed with clean seawater, placed separately in a clean plastic bag, and sealed to avoid external contamination. All the samples were stored in ice bags at low temperatures and transported to the laboratory at −4 °C for analysis as soon as possible. All the samples were collected in duplicate.

### 2.4. Sample Preparation

Sample preparation was carried out according to the methods of Tuner et al., and the steps are summarized as follows [7]. Each shellfish sample was approximately 2.5 kg in weight. The sample was washed and shaken. After the silt inside the shellfish was washed away, the edible flesh tissues (including digestive glands) were removed and subsequently homogenized. A total of 2.0 g of homogenized tissue was weighed into a 10 mL centrifuge tube, and 8 mL of 0.5% (*v*/*v*) acetic acid aqueous solution was added. The mixture was vortexed for 90 s and then extracted by ultrasonication for 5 min. The centrifuge tube was then sealed in boiling water for 5 min and cooled to room temperature quickly. The extract was centrifuged at 10,000 rpm for 5 min and prepared for purification. 

An aliquot of 1.0 mL of the supernatant was decanted into a 2.0 mL centrifuge tube, 5 μL of ammonium hydroxide was added, and the mixture was mixed well. A solid-phase extraction (SPE) procedure using Supelco ENVI-Carb cartridges (250 mg/3 mL, Supelco) was used for shellfish extract purification before the chromatographic analysis. The stationary phase was conditioned with 3.0 mL of aqueous acetonitrile (20%, *v*/*v*), which contained acetic acid (0.8%, *v*/*v*) and 3.0 mL of ammonium hydroxide (0.1%, *v*/*v*). An aliquot of 250 μL of extract was loaded onto the SPE column. The stationary phase was then rinsed with 700 μL of ultrapure water and drained for 5 s. The toxins were subsequently eluted with 2 mL of aqueous acetonitrile (20%, *v*/*v*) containing 0.8% of acetic acid (*v*/*v*) and drained under a vacuum for 5 s. The eluent was collected in a 5 mL centrifuge tube, mixed, and filtered through a 0.22 μm syringe membrane filter prior to chromatographic analysis. 

### 2.5. LC–MS Analysis

The purified sample extracts and standards were analyzed using a HILIC–LC–MS/MS [7]. An 8060 LC–MS triple quadrupole mass spectrometer (Shimadzu, Kyoto, Japan) coupled with an LC-30AD ultrahigh-performance liquid chromatography system (Shimadzu, Kyoto, Japan) was used. An XBridge Amide column (Lot No.0163370685, 3.0 mm × 150 mm, 1.7 µm; Waters, Milford, MA, USA) was used for analysis. Mobile phase A was 10 mmol/L formic acid and 5 mmol/L ammonium formate, and mobile phase B was acetonitrile. The linear gradient program was performed as follows: 0–0.01 min, 85% B; 0.01–5 min, 85% to 45% B; 5–10 min, 45% B; 10–10.5 min, 45% to 85% B; and 10.5–14 min, 85% B. The injection volume was 5 μL at a flow rate of 0.4 mL/min for the mobile phase, and the column temperature was set at 40 °C.

The tandem mass spectrometer equipped with an electrospray source acquired masses in fast polarity switching (negative and positive). Mass acquisition was carried out in multiple reaction monitoring (MRM) mode. The electrospray ion source parameters were as follows: interface voltage 4500 V; desolvent line; MS interface and heat block temperatures 250 °C, 300 °C and 400 °C, respectively; heating gas (air, 10 L/min); nebulizing gas (nitrogen, 3 L/min); drying gas (nitrogen, 10 L/min); and collision gas (argon, 270 kPa). The MS parameters are listed in Table 2.

### 2.6. Analytical Validation

The LC–MS/MS method was validated in the laboratory, and the validation parameters of the method are presented in Table 3. The limits of detection (LODs) were 5–20 µg/kg according to this method (Table 2). The chromatogram of PST is shown in Figure 2. The precision and accuracy of the method satisfied the quality-control requirements of this study.

### 2.7. Acute Dietary Exposure Risk Assessment for PSTs from Shellfish for Chinese General Consumers

#### 2.7.1. Shellfish Consumption Data of Chinese General Consumers

Shellfish consumption data were obtained from the China National Food Consumption Survey, which was conducted in 2018–2020 [22]. This survey used multistage random cluster sampling and was conducted in 18 provinces (12 coastal and 6 inland provinces) in China. Food consumption was recorded using three discontinuous, face-to-face interviews of 24 h diet recalls that were performed on one weekend day (Saturday or Sunday) and two weekdays, and two adjacent surveys of the three interviews were conducted at least five days apart. In this survey, a total of 55,700 participants, including 3200 shellfish consumers aged 3 to 92 years were surveyed. The shellfish commonly consumed included scallops, mussels, oysters and blood clams (Table 4).

#### 2.7.2. The Equivalent Concentrations of STX in Shellfish Samples

The toxicity equivalence factors (TEFs) of STX and its analogs, published by JECFA in 2016, were used to define the toxicity ratio of a compound from a chemical group that shares the same mode of action as a reference compound in the same group [20]. The toxicity of the analog was expressed as a fraction of the toxicity of the reference compound [20]. Table 5 presents the TEFs published by JECFA. In this study, the total toxin group of STX and its analogs in the shellfish samples were expressed in μg STX.2HCL equivalents (eq.) per kg shellfish tissue, using the TEFs published by JECFA. 

#### 2.7.3. Probabilistic Model of Acute Dietary Exposure to PSTs from Shellfish among Chinese General Consumers

Monte Carlo (MC) simulations were used for the probabilistic modeling of acute dietary exposure [23]. Daily shellfish consumption was simulated by sampling from the food consumption database combined with a random sample from an empirical STX equivalent concentration distribution for each shellfish. Due to the limited sample size of shellfish in this study, a *lognormal* distribution was used to fit the STX equivalent concentrations in the shellfish samples [24]. The daily acute dietary exposure to PSTs (μg STX.2HCl eq./kg bw) from shellfish was calculated by summing the intake of shellfish consumed in a single day, adjusted by the measured individual body weight of each consumer. The shellfish consumption and STX equivalent concentration data were harmonized by coding food according to the Codex Classification of Foods and Animal Feeds promoted by the Codex Alimentarius Commission (CAC) [25,26]. Based on the WHO-recommended approach for undetected values, the worst case was used to conservatively estimate acute dietary exposure to PSTs by shellfish consumers; i.e., the undetected values were assumed to be LODs [27].

The number of MC iterations was 100,000 for each group of the population to be assessed. The acute dietary exposures were specified at the percentiles P50, P75, P90, P99 and P99.9 and at the mean value of the intake distribution. Bootstrap resampling was used to evaluate the uncertainty of the MC variation in consumers and compound concentrations quantitatively, and relevant statistics (such as P50, P95 and P99) and approximate confidence intervals (95% CIs) of the percentiles were calculated. The bootstrap sample in this study was B = 100.

The results of daily acute dietary exposures were subsequently compared with the ARfD of STX (0.5 μg STX.2HCl eq./kg bw) established by the EFSA, which estimated the probability of consumers with acute health risks [18].

### 2.8. Estimation of Acute Dietary Exposure to PSTs from Shellfish by Consumers in Coastal Areas of China

Given that PST poisoning mainly occurs in coastal areas, a specific survey of shellfish consumption was carried out in coastal areas in China to assess the acute health risk of PSTs to typical consumers.

#### 2.8.1. Shellfish Consumption Data of Special Consumers in the Bohai Rim Region

A specific survey of shellfish consumption in the Bohai Rim region in China was conducted in 2019 [28]. This survey involved random sampling in local main aquatic product markets, communities, fishing villages and large factories. The consumption frequency, consumption and cooking habits of shellfish were recorded via face-to-face interviews with diet recalls of the month with the highest shellfish consumption. In this survey, a total of 875 participants aged 13 to 91 years were shellfish consumers, including 481 males and 394 females, representing the consumers who frequently consumed shellfish in the Bohai Rim area_._ The detailed consumption data for consumers are shown in Table 6.

#### 2.8.2. Probabilistic Model of Acute Dietary Exposure to PSTs from Shellfish Consumed by Typical Consumers

The acute dietary exposure of shellfish consumers to PSTs from shellfish in coastal areas of China was calculated probabilistically using *@Risk* software (version 8.2, Palisade). The probabilistic modeling used in this study was implemented by Monte Carlo simulations, which simulate the daily consumption by sampling from the shellfish consumption distribution and combining these data with a random sample from the distribution of equivalent concentrations of PSTs. Random sampling from the concentration distribution was performed according to the percentage of the sample with detectable concentrations and the percentage of the sample with undetectable concentrations. Each consumer’s acute dietary exposure to PSTs from shellfish (μg STX.2HCl eq./kg bw) was calculated by shellfish consumption combined with the equivalent concentrations of PSTs in shellfish on a single day, adjusted by the measured individual body weight of each consumer. 

*@Risk* software was used to fit the equivalent concentration data, shellfish consumption data and body weight data to obtain the appropriate distribution: the *lognorm* distribution for the equivalent concentrations of STX in shellfish samples above the LOD. Considering the uncertainty and variability in the concentrations of the undetected samples, the equivalent concentrations of PSTs in the undetected samples were assumed to have a distribution ranging from 0 to the maximum LOD for PSTs, also expressed as the equivalent concentrations of PSTs. A *Uniform* distribution was assumed for the equivalent concentrations of PSTs of the LOD in the case of undetectable samples. The *Pert* distribution was applied to consumption data. The *Normal* distribution was used for shellfish consumers’ body weights. The details of the variables and models used for this acute exposure assessment are shown in Table 7. The number of Monte Carlo iterations was 100,00, and the number of simulations was 100. The exposures were specified at percentiles P50, P90, P95, P97.5 and P99 and the mean from the intake distribution and compared with the ARfD of the STX.

### 2.9. Statistical Analysis

The data were analyzed using SPSS (version 25.0 for Windows, Armonk, NY: IBM Corp., USA). The PST levels in the samples are shown as the minimum to maximum values of the detected PSTs. The acute dietary exposure of Chinese general consumers to PSTs from shellfish was statistically analyzed via SAS software (version 9.4). *@Risk* software was used to carry out acute dietary exposure assessments of PSTs for typical consumers in coastal areas of China.

## 3. Results and Discussion

### 3.1. PST Toxin Content in Shellfish in the Yellow-Bohai Sea in Dalian

The contents of PSTs in common bivalve and gastropod (such as veined rapa whelks) samples from the Yellow-Bohai Sea in Dalian were analyzed. As shown in Table 8, the highest detection rate of PSTs was found in scallops and blood clams (both 94.4%), followed by mussels (20.0%). Notably, PSTs were not detected in Manila clams or veined rapa whelks. The maximum levels of total PSTs were 159.3, 235.8, 3953.5 and 993.4 μg STX.2HCl eq./kg in pacific oysters, blue mussels, scallops and blood clams, respectively. Among all the shellfish species, the detection rate and content of PSTs in scallops were greater than those in other shellfish species, followed by those in blood clams. The highest contents of STX, neoSTX, GTX1&4, GTX2&3, GTX5, dcSTX and dcGTX2&3 in the scallop samples were recorded at 928.3, 893.6, 1409.3, 259.6, 558.6, 359.4, 39.4, 95.9, 77.6 and 34 μg STX.2HCl eq./kg, respectively. The highest contents of STX, neoSTX, GTX1&4, GTX2&3, GTX5, dcSTX and dcGTX2&3 in blood clam samples were recorded at 235.8, 309.0, 238.5, 69.4, 179.4, 84.4, 10.3, 29.9, 43.7 and 18.9 μg STX.2HCl eq./kg, respectively. This was related to the fact that mussels can rapidly accumulate and eliminate PSTs and that scallops do so relatively slowly, while compared to scallops and mussels, oysters and clams are less able to accumulate PSTs [29,30]. In addition, the sampling period, exposure time to toxin algae and the metabolic capacity of shellfish, were also influencing factors [8].

Due to the common consumption pattern of scallop viscera by consumers in coastal cities in China, this study divided each scallop sample into three parts: the scallop adductor muscle, the scallop skirts, and the scallop viscera, which were subsequently individually analyzed. The PST levels in different parts of the scallop samples are shown in Table 9. The highest detection rate of PSTs in scallop viscera was 94.4%, followed by 79.4% in scallop skirts, and 29.4% in scallop muscles. The maximum levels of total PSTs in the scallop adductor muscle, the scallop skirts and the scallop viscera were 332.1, 1015.9 and 2605.5 μg STX.2HCl eq./kg, respectively. Therefore, the consumption of viscera was the most significant risk factor for PST poisoning [31].

### 3.2. Seasonal Variation in the PSTs in Bivalves and Gastropods in the Yellow-Bohai Sea in Dalian

One-year monitoring of PSTs was conducted on common bivalves and gastropods in the Yellow-Bohai Sea in Dalian. This monitoring of PSTs focused on the aquaculture process in this study. The contamination levels and detection rates of PSTs in shellfish samples throughout the year are shown in Figure 3. The overall detection rate of PSTs in shellfish samples reached its peak in September (50%). Moreover, the average contamination level of PSTs in bivalves also reached its peak for the entire one-year monitoring period in September 2020. In addition, July was the month with the lowest PSTs during the one-year continuous monitoring period from 2019 to 2020.

Seasonal differences in the contamination levels of PSTs were observed in bivalves during aquaculture in the Dalian coastal area of the Yellow and Bohai Seas, while the seasonal variations in PSTs in different sea regions also differed. In Zhou’s study (from March 2019 to February 2020), the highest PST levels in shellfish samples were March > December > January in Shenzhen City’s Buji seafood market [9]. Similarly, another study showed that the detection rates of PSTs were high from April to June in Bohai Bay in Tangshan, China, with a maximum PST level of 532.57 μg STX.2HCl eq./kg in oysters in April 2020 [32]. Liu et al. found that phytoplankton samples collected in June and September in the Bohai Sea mariculture zone in China had higher PST levels than samples collected in November and December, suggesting the possible proliferation of toxic algae in spring and autumn [33]. In this study, autumn was the season of concern for PST contamination and seafood poisoning in the Yellow-Bohai Sea in Dalian. The seasonal and regional differences in PSTs are mainly due to the differences in the dominant algal species in different sea areas [34]. Additionally, these differences are influenced by factors such as toxicity, the abundance of toxic microalgae in the seawater, and the duration of exposure to these harmful algae [35]

### 3.3. Probabilistic Estimation of Acute Dietary Exposure to PSTs for Chinese General Consumers

The results of acute dietary exposure to PSTs for Chinese general consumers are shown in Table 10. For general consumers, the average acute dietary exposure to PSTs from shellfish was 0.27 μg STX.2HCl eq./kg bw. The average acute dietary exposure to PSTs from shellfish in different age groups ranged from 0.22 to 0.46 μg STX.2HCl eq./kg bw. At the P97.5, acute dietary exposure was estimated to be 1.24 μg STX.2HCl eq./kg bw for general consumers, 2.37 μg STX.2HCl eq./kg bw for children aged 3–6 years, 1.86 μg STX.2HCl eq./kg bw for adolescents aged 7–12 years, 1.28 μg STX.2HCl eq./kg bw for adolescents aged 13–18 years,1.10 μg STX.2HCl eq./kg bw for adults and 0.94 μg STX.2HCl eq./kg bw for people aged more than 60 years old. In addition, among general consumers, the probability of acute health risks to shellfish consumers from dietary exposure to PSTs was 13.14% (Figure 4). The probability of acute health risks to shellfish consumers in different age groups from dietary exposure to PSTs ranged from 9.86% to 26.97%, with the highest probability occurring in children aged 3–6 years. Therefore, general consumers, especially children aged 3–6 years, and consumers with high shellfish consumption should be concerned during the occurrence of harmful algal blooms.

### 3.4. Probabilistic Estimation of Acute Dietary Exposure to PSTs for Typical Consumers in Coastal Areas of China

The results of the probabilistic assessment are shown in Table 8. For scallop consumers, the average and high acute exposures to PSTs from scallops were 0.64 and 2.71 μg STX.2HCl eq./kg bw (P97.5), respectively. Among typical scallop consumers, the probability of acute health risk from dietary exposure to PSTs was 40.8% (Figure 5A). However, if scallop consumers avoided eating scallop viscera, the average acute exposure to PSTs from scallops was 0.41 μg STX.2HCl eq./kg bw, and the probability of acute health risk to shellfish consumers from dietary exposure to PSTs was 27.2% (Figure 5B). For blood clam consumers, the average and high acute exposures to PSTs from blood clams were 0.67 and 2.31 μg STX.2HCl eq./kg bw (P97.5), respectively. Among frequent blood clam consumers, the probability of acute health risk from dietary exposure to PSTs was 49.2% (Figure 5C). For consumers of other shellfish (mussels and oysters), the average and high acute exposures to PSTs from other shellfish were 0.36 and 1.60 μg STX.2HCl eq./kg bw (P97.5), respectively. Among frequent mussel and oyster consumers, the probability of acute health risk to shellfish consumers from dietary exposure to PSTs was 22.1% (Figure 5D). In this study, for typical consumer populations in coastal areas in China, especially those with higher shellfish intake or those who eat scallop viscera, there was a certain health risk associated with acute exposure to PSTs through bivalve consumption during the occurrence of harmful algal blooms.

Studies of acute dietary exposure assessment of PSTs in bivalves have been conducted in several countries and other coastal regions of China. According to the EFSA report, the acute dietary exposure to PSTs was 4.3 μg STX.2HCl eq./kg bw, which was calculated based on the P95 contamination level of PSTs in bivalve samples in the market and the consumption of 400 g/day for adults [18]. A study in Korea showed that the acute dietary exposure to PSTs was 0.3 μg STX.2HCl eq./kg bw calculated based on the maximum contamination level of PSTs in bivalve samples and the P95 of shellfish consumption (88 g/d) [10]. Additionally, an acute dietary exposure assessment of PSTs in Shenzhen residents in China showed that the PST exposure of scallop consumers ranged from 1.65 to 1.84 μg STX.2HCl eq./kg bw, and there was a risk of acute poisoning by PSTs [9]. The differences in exposure levels may be due to the point assessment method used in both the studies and differences in the contamination levels of PST and consumption of shellfish in different countries. 

### 3.5. Uncertainty Analysis

Some uncertainties may have influenced the estimated exposures in this study. Firstly, the types of sample monitoring did not fully cover all types of seafood that PSTs could have contaminated, mainly involving the common bivalves. Secondly, the HILIC–LC–MS/MS method was used to analyze the main PST toxins in the shellfish in this study, but not all PSTs (known and unknown structures) were included, and the results may be underestimated. Meanwhile, samples with levels below the LOD were assumed to be at the LOD in the exposure assessment, which might overestimate PST exposure from shellfish. Thirdly, a questionnaire survey was used to recall the consumption of shellfish in one month when consumers consumed the highest amount of seafood, and there was uncertainty between the maximum amount of shellfish consumption recalled by consumers and the actual maximum consumption in the specific survey of shellfish consumption. Furthermore, in this study, the content data of PSTs in 199 shellfish samples were used to estimate the acute dietary exposure to PSTs in Chinese general consumers, and the representativeness of the shellfish sample size may bring uncertainty to the results. Finally, due to the highly skewed distribution of levels of PSTs, the high-percentile exposure still varied under the condition of more MC simulations, and there was uncertainty (Appendix A).

## 4. Conclusions

The current study determined the PST levels in shellfish from 2019 to 2020 in the Dalian area from the Yellow-Bohai Sea using the HILIC–LC–MS/MS method and estimated their potential acute health risks to general Chinese consumers and typical consumers in coastal areas of China. Among the samples, scallops and blood clams were the shellfish species with the highest detection rate of PSTs (94.4%), and the highest total content of PSTs was detected in scallops (3953.5 μg STX.2HCl eq./kg), followed by blood clams (993.4 μg STX.2HCl eq./kg). The highest detection rate of PSTs was in the scallop viscera (94.4%), and the maximum level of total PSTs was in the scallop viscera (2605.5 μg STX.2HCl eq./kg). Consumption of scallop viscera was the most significant risk factor for PST poisoning. In addition, autumn was the season of concern for PST contamination and seafood poisoning in the Yellow-Bohai Sea in Dalian. For the Chinese general consumers, the probability of acute health risks to shellfish consumers from dietary exposure to PSTs was 13.14%; for typical consumers in the coastal areas of China, especially those with higher shellfish intake, there was an acute health risk associated with exposure to PSTs through bivalve consumption during the occurrence of harmful algal blooms. Therefore, it is suggested that the government continue to strengthen the monitoring of the source of PSTs and the monitoring of harmful algal blooms and reduce the production of industrial and domestic wastewaters, as well as give reasonable advice on shellfish consumption for the consumers in coastal areas, such as not eating scallop viscera.

## Figures and Tables

**Figure 1 foods-13-00361-f001:**
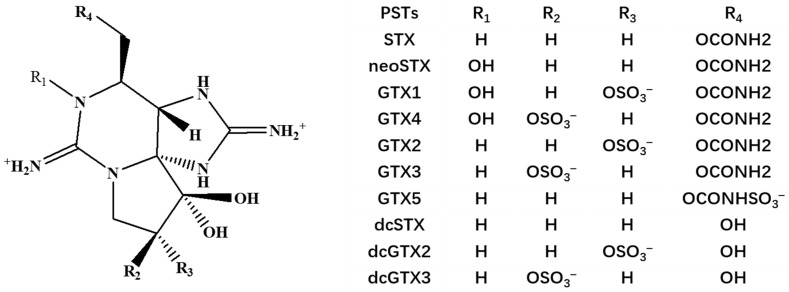
Chemical structures of PSTs.

**Figure 2 foods-13-00361-f002:**
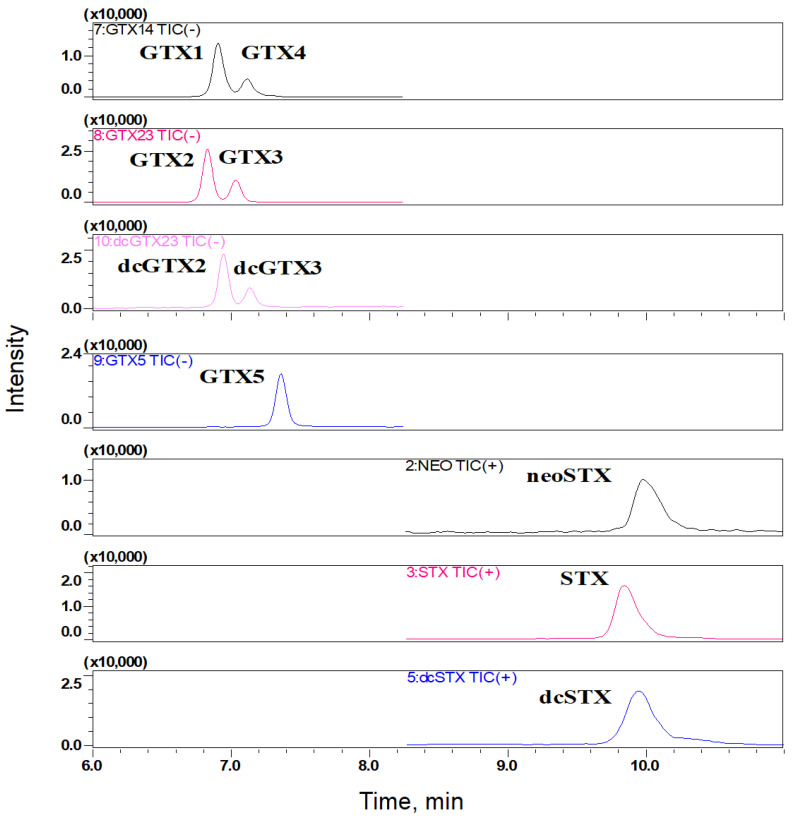
Chromatogram obtained by the analysis of PST standard solution by HILIC–LC–MS/MS following the conditions described in Table 2 and Table 3.

**Figure 3 foods-13-00361-f003:**
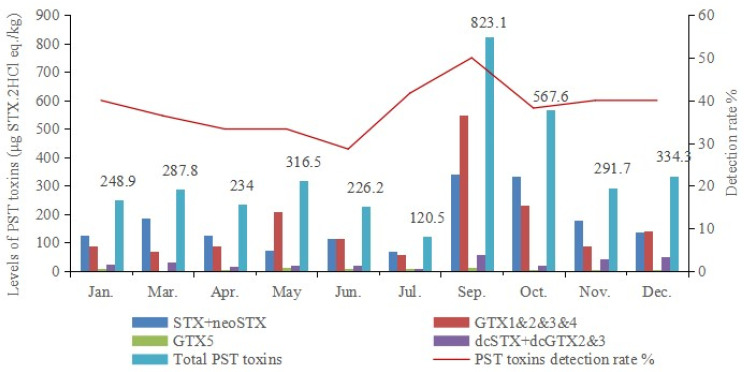
The levels and detection rates of PSTs in shellfish samples from 2019 to 2020 in the Yellow-Bohai Sea in Dalian.

**Figure 4 foods-13-00361-f004:**
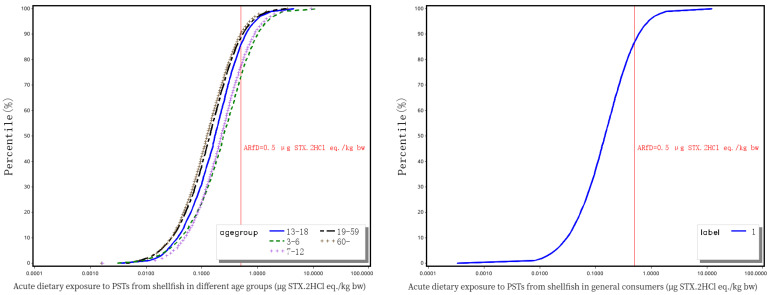
The proportions of general shellfish consumers exceeding the ARfD of the STX for Chinese consumers.

**Figure 5 foods-13-00361-f005:**
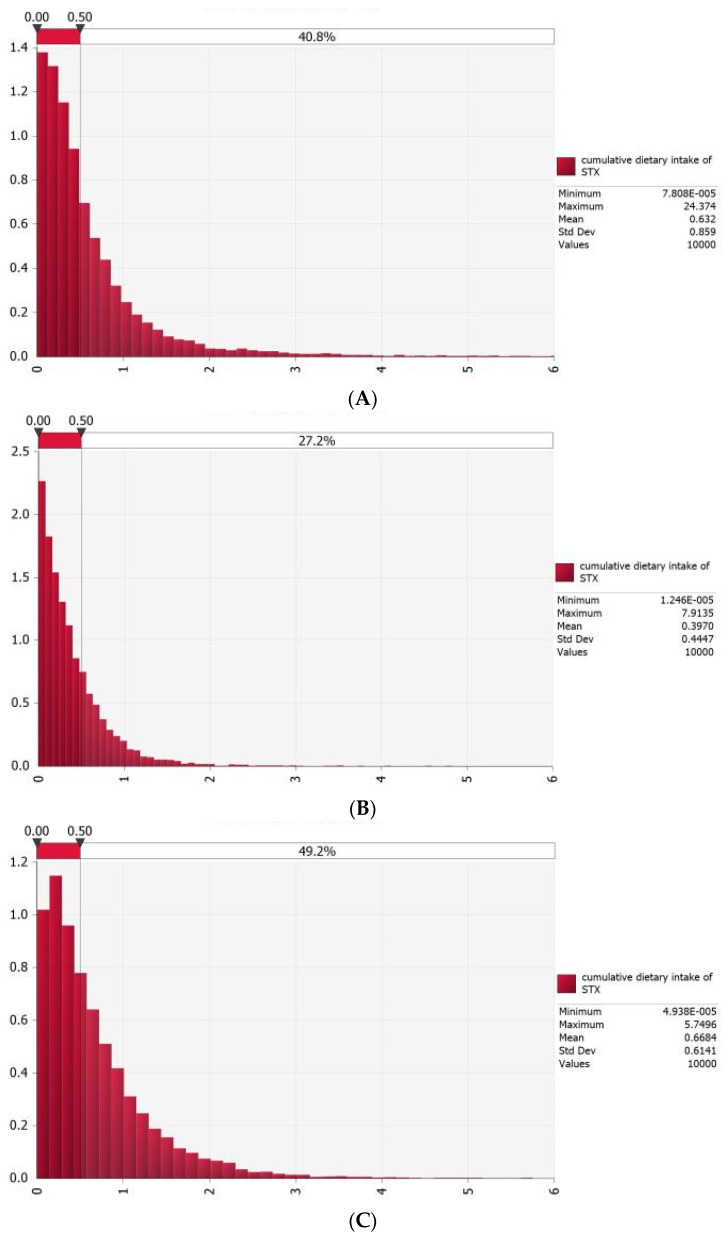
The probability of acute health risk from dietary exposure to PSTs in typical scallop consumers (**A**), scallop consumers avoided eating scallop viscera (**B**), frequent blood clam consumers (**C**) and consumers of other shellfish (mussels and oysters) (**D**) in coastal areas of China.

**Table 1 foods-13-00361-t001:** The standard solution preparation.

Toxins	Molar Concentration (μmol/L)	Mass Concentration (μg/mL)	Standard Solution Volume (μL)	Blank Matrix Volume (μL)	Total Volume (μL)	The Concentrations of Mixed Standard Intermediate Solutions (ng/mL)
STX	66.3	24.68	20	1840	2000	247
neoSTX	65.6	25.47	20			255
dcSTX	65	21.40	20			214
GTX5	55.7	25.19	40			504
GTX1	60.4	29.25	25			366
GTX4	19.7	9.54				119
GTX2	114.2	53.48	15			401
GTX3	43.4	20.32				152
dcGTX2	116	49.33	20			493
dcGTX3	26.1	11.10				111

**Table 2 foods-13-00361-t002:** MS acquisition parameters for PSTs.

Toxin	ESI Mode	Precursor Ion (*m*/*z*)	Collision Energy (eV)	Product Ion (*m*/*z*)	Collision Energy (eV)	LOD (µg/kg)
STX	ESI^+^	300.2/204.0	30	300.2/138.1	35	5
dcSTX	ESI^+^	257.1/126.1	30	257.1/221.1	22	5
neoSTX	ESI^+^	316.1/298.1	34	316.1/220.1	34	20
GTX2	ESI^-^	394.0/351.1	16	394.0/333.1	22	10
GTX3	ESI^-^	394.0/333.1	22	394.0/351.1	16	10
GTX1	ESI^-^	410.1/367.1	15	410.1/349.1	22	10
GTX4	ESI^-^	410.1/349.1	22	410.1/367.1	15	10
GTX5	ESI^-^	378.1/122.1	25	378.1/360.1	20	10
dcGTX2	ESI^-^	351.1/164.0	30	351.1/333.1	17	20
dcGTX3	ESI^-^	351.1/333.1	17	351.1/164.0	30	20

**Table 3 foods-13-00361-t003:** The recoveries of PSTs in mussel, scallop and oyster.

Shellfish Species	Toxins	Standard Solution Concentration (μg/kg)	Recovery (%)	RSD_r_ (%)
Mussel	STX	98.8	101.0	11.6
	neoSTX	102.0	111.0	8.9
	dcSTX	85.6	92.1	9.0
	GTX5	202.0	109.0	10.4
	GTX1	146.0	86.3	8.1
	GTX4	47.6	82.0	13.9
	GTX2	160.0	82.9	8.3
	GTX3	60.8	86.3	12.8
	dcGTX2	197.0	98.5	7.0
	dcGTX3	44.4	86.3	10.4
Scallop	STX	198.0	94.8	6.8
	neoSTX	204.0	102.0	9.0
	dcSTX	171.0	91.6	3.0
	GTX5	403.0	111.0	5.8
	GTX1	293.0	102.0	5.9
	GTX4	95.2	106.0	9.8
	GTX2	321.0	118.0	10.6
	GTX3	122.0	109.0	14.0
	dcGTX2	394.0	114.0	10.8
	dcGTX3	88.8	112.0	11.3
Oyster	STX	494.0	101.0	8.0
	neoSTX	510.0	107.3	7.3
	dcSTX	428.0	97.1	4.7
	GTX5	1008.0	101.1	9.6
	GTX1	732.0	98.5	8.0
	GTX4	238.0	88.5	8.0
	GTX2	802.0	101.3	9.0
	GTX3	304.0	99.4	9.8
	dcGTX2	986.0	101.9	6.8
	dcGTX3	44.4	104.2	5.0

**Table 4 foods-13-00361-t004:** Food consumption data of shellfish in the China National Food Consumption Survey.

Shellfish Species	Consumption Days	Consumption of Shellfish (g/d)					
		Mean	P50	P90	P97.5	P99	Max
Manila clam	2153	88.4	75	180	250	300	500
Scallop	471	42.3	20	100	250	500	650
Mussel	112	78.4	75	150	250	500	500
Oyster	1154	61.1	50	110	200	260	650
Blood clam	9	115.6	110	200	200	200	200

**Table 5 foods-13-00361-t005:** The toxicity equivalence factors (TEFs) for PSTs.

STX and Analogues	TEFs
STX	1.0
neoSTX	2.0
GTX1	1.0
GTX2	0.4
GTX3	0.6
GTX4	0.7
GTX5	0.1
dcSTX	0.5
dcGTX2	0.2
dcGTX3	0.4

**Table 6 foods-13-00361-t006:** Consumption data of shellfish in Bohai Rim area in China.

Shellfish Species	Number of Consumers	Proportion of Consumers (%)	Average Monthly Consumption Frequency	Consumption of Shellfish (g/d/Person)			
				Mean	P50	P97.5	Range
Scallop	737	84.2	4	183.8	175.0	525.0	10.2~875.0
Mussel	432	49.4	2	302.0	245.0	980.0	24.5~1225.0
Oyster	672	76.8	3	191.1	125.0	625.0	20.0~1875.0
Blood clam	184	21.0	1	114.0	87.5	375.0	8.8~500.0

**Table 7 foods-13-00361-t007:** Description and distribution of variables and models for acute dietary exposure assessment of PSTs in consumers.

Variables	Definition	Assumption/Distribution/Formula	Source
Conc_STX_ ***	Equivalent concentrations of PSTs in detected samples	*Lognorm* (120.67, 223.77) ^a^	Monitoring data
		*Lognorm* (326.75, 143.34) ^b^	
		*Lognorm* (285.39, 136.24) ^c^	
Conc_STX_ ****	Equivalent concentrations of PSTs in undetected samples	*Uniform* (0, 118.5)	Monitoring data
P_p_	Rate of shellfish sample with detectable concentration	94.4% ^a^	Calculated
		94.4% ^b^	
		12.5% ^c^	
P_n_	Rate of shellfish sample with undetectable concentration	5.6% ^a^	Calculated
		5.6% ^b^	
		87.5% ^c^	
Conc_STXs_	Equivalent concentrations of PSTs in shellfish samples	*Discrete* (Conc_STX_ *:Conc_STX_ **, P_n_:P_p_)	Calculated
Cons	Consumption of shellfish in Bohai Rim region in China	*Pert* (0, 61.4, 979.3) ^a^	Specific survey of shellfish consumption in Bohai Rim region
		*Pert* (8.5, 29.5, 588.7) ^b^	
		*Pert* (20, 20, 1891.7) ^c^	
bw	Body weight for shellfish consumers in Bohai Rim region in China	*Normal* (70.3, 13.1)	Specific survey of shellfish consumption in Bohai Rim region

Note: * indicates the equivalent concentrations of PSTs in detected samples; ** indicates the equivalent concentrations of PSTs in undetected samples. ^a^ The scallop; ^b^ The blood clam; ^c^ The mussel and oyster.

**Table 8 foods-13-00361-t008:** PST levels in shellfish samples from the Yellow-Bohai Sea in Dalian in China.

Shellfish Species	Samples	Detection Rate of PSTs (%)	PSTs in Detected Samples (μg STX.2HCl eq./kg)										
			STX	neoSTX	GTX1	GTX4	GTX2	GTX3	GTX5	dcSTX	dcGTX2	dcGTX3	Total Content of PSTs
Pacific oyster	34	5.9	11.3	ND	36.2~44.7	10.7~14.0	38.9~43.3	23.9~25.7	1.8~1.9	ND	11.6~12.8	6.2~8.9	132.6~159.3
Mediterranean blue mussels	25	20.0	15.2~235.8	ND	118.8~149.9	25.9~32.1	5.4~19.1	13.3~14.2	ND	ND	ND	ND	5.4~235.8
Manila clam	34	0	ND	ND	ND	ND	ND	ND	ND	ND	ND	ND	ND
Scallop	36	94.4	5.9~928.3	23.2~893.6	12.9~1409.3	7.0~259.6	4.8~558.6	6.1~359.4	1.0~39.4	5.7~95.9	4.9~77.6	5.2~34.0	43.0~3953.5
Veined rapa whelks	34	0	ND	ND	ND	ND	ND	ND	ND	ND	ND	ND	ND
Blood clam	36	94.4	12.2~235.8	20.6~309.0	10.1~238.5	5.0~69.4	8.9~179.4	6.5~84.4	2.2~10.3	6.5~29.9	3.8~43.7	4.1~18.9	94.4~993.4
Total	199	37.3	5.9~928.3	20.6~893.6	10.1~1409.3	5.0~259.6	4.8~558.6	6.1~359.4	1.0~39.4	5.7~95.9	3.8~77.6	4.1~34.0	5.4~3953.5

Note: the scallop adductor muscle, scallop skirts and scallop viscera were analyzed for each scallop sample. ND indicates not detected.

**Table 9 foods-13-00361-t009:** PST levels in different parts of scallops from the Yellow-Bohai Sea in Dalian.

Scallops	Detection Rate of PSTs (%)	PSTs in Detected Samples (μg STX.2HCl eq./kg)										
		STX	neoSTX	GTX1	GTX4	GTX2	GTX3	GTX5	dcSTX	dcGTX2	dcGTX3	Total Content of PSTs
Scallop adductor muscle	29.4	7.6~96.8	42.8~79.0	24.9~67.3	8.3	4.4~49.9	6.7~18.8	1.3~7.4	ND	8.3	ND	4.4~332.1
Scallop viscera	94.4	5.9~397.1	23.2~893.6	12.9~1145.0	7.0~182.6	4.8~374.6	3.1~239.2	1.0~24.8	6.6~57.0	2.9~49.8	5.2~33.8	10.0~2605.5
Scallop skirt	79.4	6.9~434.4	32.0~58.2	14.7~197.0	7.3~68.7	4.4~134.1	6.2~101.3	1.3~14.0	5.7~38.9	3.7~19.5	6.9~9.2	10.0~1015.9

Note: ND indicates not detected.

**Table 10 foods-13-00361-t010:** Probabilistic estimation results of acute dietary exposure to PSTs from shellfish in Chinese general consumers.

Group	Acute Dietary Exposure to PSTs from Shellfish (μg STX.2HCl eq./kg bw)								QR	95% CI
	Mean	P25	P50	P75	P90	P97.5	P99	P99.9		
3~6	0.46	0.10	0.25	0.55	1.07	2.37	3.16	6.73	0.10	2.01~2.37
7~12	0.40	0.11	0.23	0.47	0.88	1.86	2.69	5.77	0.07	1.64~1.92
13~18	0.29	0.08	0.17	0.35	0.61	1.28	1.92	4.38	0.08	1.19~1.38
19~59	0.24	0.06	0.14	0.29	0.54	1.10	1.60	3.28	0.01	1.09~1.14
≥60	0.22	0.06	0.13	0.26	0.48	0.94	1.49	3.38	0.03	0.90~0.99
Total	0.27	0.07	0.15	0.31	0.60	1.24	1.85	3.84	0.02	1.22~1.26

Note: P, percentile; QR, interquartile range of P97.5; 95% CI, confidence interval of P97.5.

## Data Availability

The data presented in this study are available on request from the corresponding author.

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
