# Peer review of "Contamination Status and Acute Dietary Exposure Assessment of Paralytic Shellfish Toxins in Shellfish in the Dalian Area of the Yellow-Bohai Sea, China"

_foods, 2024, doi:10.3390/foods13030361_

Round 1

Reviewer 1 Report

Comments and Suggestions for Authors

The assessment of paralytic shellfish poisoning has been performed in appropriate way and the results are clearly shown. However, some critical points listed below should be improved.

1, The font of the manuscript should be unified.

2, The legends of Figure 3 are hard to read. They should be presented in bigger letters.

3,  Although the assessment is the main subject of the manuscript, the assessment is needed when the poisoning occurred. Authors should present data on historical aspect of paralytic poisoning outbreak in the Dalian area along with observed symptoms.

Comments on the Quality of English Language

Authors are advised to receive a grammar check by native speaker of English. In addition, authors should check the manuscript thoroughly for spelling mistakes.

Author Response

Comment of reviewer 1

Comment 1: The font of the manuscript should be unified.

 >Response: We agreed with the suggestion and unified the font of the manuscript as the “Times New Roman”. (See all the changes in the revised manuscript)

Comment 2: The legends of Figure 3 are hard to read. They should be presented in bigger letters.

>Response: We agreed with the suggestion, and revised the legends of Figure 3 in the revised manuscript. ï¼ˆSee the revised Figure 4)

Comment 3: Although the assessment is the main subject of the manuscript, the assessment is needed when the poisoning occurred. Authors should present data on historical aspect of paralytic poisoning outbreak in the Dalian area along with observed symptoms.

>Response: In accordance with the reviewer’s comments, we added the details of the poisoning of PST in Fujian Province in 2017, including the symptoms, the number of consumers poisoned and incubation period in the revised manuscript. However, there was currently lack of the published report of PST poisoning in Dalian, we added the description of contamination of PST in shellfish in Dalian, Liaoning Provence in the revised manuscript. (See line 74-77 in the revised manuscript) 

Comment 4: Authors are advised to receive a grammar check by native speaker of English. In addition, authors should check the manuscript thoroughly for spelling mistakes.

>Response: The second reviewer has put forward modification suggestions for many errors in English expression in the manuscript. We have carefully checked and modified the English expression according to the reviewer's comments. See all parts marked in yellow in the revised manuscript.

Reviewer 2 Report

Comments and Suggestions for Authors

The PDF document of the manuscript contains all the observations and corrections recommended in this review process. The manuscript needs extensive revision and reorganization. The material is within the scope of the publisher.

Author Response

Comment of reviewer 2

Comment 1: Please include PST chemical structure as figure in the manuscript.

>Response: We agreed with the suggestion, and added the Figure of PST chemical structure in the revised manuscript. ï¼ˆSee the Figure 1 in the revised manuscript)

Comment 2: It is necessary to provide some chromatographic result (chromatogram), as well as a table summarizing all analytical parameters resulting from the validation of the HILIC-LC-MS/MS method.

>Response: We agreed with the suggestion, and added the chromatogram of PST and validation parameters in the revised manuscript. ï¼ˆSee the Figure 2 and Table 3 in the revised manuscript)

Comment 3: The introduction needs a very significant improvement.

>Response: We agreed with the suggestion, and revised the Introduction in the revised manuscript. ï¼ˆSee the line 40-107 in the revised manuscript)

Comment 4: Section 2.2: Put all this concentration values in a table.

>Response: We agreed with the suggestion, and revised it in the revised manuscript. ï¼ˆSee the Table 1 in the revised manuscript)

Comment 5: Review the sample collection procedures, treatment, as well as description of the procedures.

>Response: We agreed with the suggestion, and revised it in the revised manuscript. ï¼ˆSee  the line 128-149 in the revised manuscript)

Comment 6: There are errors in the expression of results, especially the units in which they are expressed.

>Response: We agreed with the suggestion, and revised the incorrect English expressions and units. ï¼ˆSee the changes marked in yellow in the revised manuscript)

Comment 7: Use the same style format...if you type in Times New Roman keep the same style in all manuscript.

>Response: We agreed with the suggestion, and used Times New Roman in the revised manuscript. ï¼ˆSee changes in the revised manuscript)

Comment 8: In the section 2.1: All solvents used for chromatographic analysis were HPLC-Grade and chemicals such as .....were analytical grade.

>Response: We agreed with the suggestion, and revised it in the revised manuscript.(See the line 113-116 in the revised manuscript)

Comment 9: Authors used conditions described by Tunner et al, please use this reference, If a special SOPO were used please report the reference in bibliography. Which SOP the authors use? Is a internal document? All this references and information should be clearly report.

>Response: We agreed with the suggestion, and added this reference in the revised manuscript. ï¼ˆSee the line 151-152 in the revised manuscript)

Comment 10: Column batch number should be report, This stationary phase are not robust, is weakly stable, for this reason batch number is required in order to get traceability of the column.

>Response: We agreed with the suggestion, and added the column batch number (Lot No. 0163370685) in the revised manuscript. ï¼ˆSee the line 181 in the revised manuscript)

Comment 11: Why authors not us Ultraturrax system? Homogenizer system is recommended.

>Response: Sample preparation was carried out with the reference described by Tunner et al, and homogenizer system was not used.

Comment 12: 10000 r/min what this mean? Remember:“Compute Relative Centrifugal Force (RCF) or G-force” or “Compute revolutions per minute (RPM)”. RPM denotes the number of revolutions a rotating object is doing per minute while the RCF denotes the force applied on an object in a rotating environment.

>Response: We agreed with the suggestion, and revised it to “10000 rpm” in the revised manuscript. ï¼ˆSee the line 161 in the revised manuscript)

Comment 13: 5μL of ammonia water, xx% (v/V) aqueous ammonia solution, for example 25%(v/V) ammonia solution?ammonium water?

>Response: We agreed with the suggestion, and revised it to “ammonium hydroxide” in the revised manuscript. ï¼ˆSee the line 168-169 in the revised manuscript)

Comment14: This work does not aim to validate the HILIC method. I believe that this method is applied, already validated in the laboratory, it is recommended that the validation parameters of the method be presented in the results in the form of a table.

>Response: We agreed with the suggestion, and added the validation parameters in the revised manuscript. ï¼ˆSee the Table 3 in the revised manuscript)

Comment 15: LOD: Should be in same unity as regulation value (regulatory level), in µgSTX.2HCLeq./Kg. The value should be reported in this unity.

>Response: LOD represented the sensitivity of the analytical method. We think it does not need to be converted to the STX equivalent concentration for LOD. However, according to the reviewer's suggestion, we use “µg STX.2HCL eq./kg” to represent SXT equivalent concentration in the manuscript.

Comment 16: STX is the reference of TEF for PST, its value is 1,0. Do you mean: PST toxicity equivalence factor were taking into account for this study and evaluation? Author must explain and reason this study and detail it well, it is not understood.

>Response: In this study, PST toxins were converted into the STX equivalent concentrations according to TEFs of STX and its analogues, published by JECFA in 2016, and used the STX equivalent concentrations in the shellfish samples in the acute dietary exposure assessment. Then the results of daily acute dietary exposures were then compared with the ARfD of STX (0.5 μg STX.2HCl eq./kg bw) established by EFSA, and accessed the acute health risks in the consumers. EFSA and JECFA also use the same method to estimate the acute dietary exposure to PST in consumers. We agreed with the suggestion, and revised the description in this section. ï¼ˆSee the line 241-282 in the revised manuscript)

Comment 17: The description of the 2.6.2 is not clear, the authors should reformulate the experiment in a way that can be clearly understood.

>Response: We agreed with the suggestion, and revised the description in this section. (See the line 241-252 in the revised manuscript)

Comment 18: Review the entire text of the description, the 2.6.5 is not understood very well, this section needs a strong revision and achieve a description that allows easy understanding of the study carried out.

>Response: In this study, according to the different shellfish consumption data available (one is the 2018-2020 China National Food Consumption survey database, and the other is the shellfish consumption survey database in typical areas) , we used the non-parametric and parametric probability assessment methods to assess the acute health risks of exposure to PST for general consumers and typical consumers, respectively. We agreed with the suggestion, and revised the description in this section.  ï¼ˆSee the line 283-337 in the revised manuscript)

Reviewer 3 Report

Comments and Suggestions for Authors

The manuscript includes an interesting study on a concrete contamination location. It is well presented and justified. My main concern regards the sampling procedure. Some more information on it ought to be provided before a subsequent revision is done.

Some concrete aspects would be as follows:

Abstract

Line 18: … 199 shellfish samples. Clarify species and number of corresponding samples.

Line 21: followed.

Keywords

Include human health.

Material and methods

Line 90: Provide details on the 199 sample distribution among species, location and catching time.

Line 241-242: This would be the error of the analytical tool. But, how many replicates of each experimental condition were carried out ? As previously mentioned, this would be my main concern.

Conclusions

Not only monitoring the source of toxic compound ought to be enhanced. Production of industrial and domestic wastewaters ought to be reduced; this aspect could also be encouraged.

General

Moderate performances of English language ought to be carried out.

Comments on the Quality of English Language

Moderate performances could be done.

Author Response

Comment of reviewer 3

Comment 1: My main concern regards the sampling procedure. Some more information on it ought to be provided before a subsequent revision is done.

>Response: We agree with the review's suggestion, and added the sampling procedures in the revised manuscript.  (See line 128-149 in the revised manuscript)

Comment 2: Line 18: … 199 shellfish samples. Clarify species and number of corresponding samples.

>Response: We agree with the review's suggestion, and added the species and number of the samples. (See line 18-20 in the revised manuscript)

Comment 3: Keywords include human health.

>Response: We agree with the review's suggestion, and added the Keywords.  (See line 39 in the revised manuscript)

Comment 4: Line 90: Provide details on the 199 sample distribution among species, location and catching time.

>Response: We agree with the review's suggestion, and added the details of the 199 samples in the revised manuscript. (See line 128-143 in the revised manuscript)

Comment 5: Line 241-242: This would be the error of the analytical tool. But, how many replicates of each experimental condition were carried out ? As previously mentioned, this would be my main concern.

>Response: We checked it, and it was a error in English expression. We revised it in the the revised manuscript. In this study, each sample was tested once, and the same batch have 10% parallel samples and labeled samples to ensure the quality of the analysis.

Comment 6: Conclusions: Not only monitoring the source of toxic compound ought to be enhanced. Production of industrial and domestic waste waters ought to be reduced; this aspect could also be encouraged.

>Response: We agree with the review's suggestion, and added it in the revised manuscript. (See line 542-543 in the revised manuscript)

Comment 7: General, moderate performances of English language ought to be carried out.

>Response: The second reviewer has put forward modification suggestions for many errors in English expression in the manuscript. We have carefully checked and modified the English expression according to the reviewer's comments. See all parts marked in yellow in the revised manuscript.

Round 2

Reviewer 2 Report

Comments and Suggestions for Authors

Some details should be revised in the manuscript, correct and imporve.

Author Response

Comment of reviewer 2

Comment 1: Please use a table, where each toxins and their chemical group can be clearly assigned.  

>Response: We agreed with the suggestion and revised the Figure 1. (See Figure 1 in the revised manuscript)

Comment 2: LC-MS/MS revised to HILIC-LC-MS/MS.

>Response: We agreed with the suggestion and revised it.(See the revised manuscript)

Comment 3: Modify the title of Table 2 and Table 3.

>Response: We agreed with the suggestion and revised them.(See the Table 2 and Table 3 in the revised manuscript)

Comment 4: Correctly should be GTX2&3 and GTX 1&4, should be grouped in identical epimers. Please correct all identical situation in this paragraph.

>Response: We agreed with the suggestion and revised it.(See the revised manuscript)

Comment 5: Please improve the resolution of the figures, I cannot understand due to low resolution the content of legend and almost total figure.

>Response: We agreed with the suggestion, and revised the Figure 5 in the revised manuscript. ï¼ˆSee the Figure 5 in the revised manuscript)

Reviewer 3 Report

Comments and Suggestions for Authors

The manuscript has been performed according to previous suggestions. I would recommend its acceptation.

Comments on the Quality of English Language

English performances during editing would be necessary.

Author Response

Comment of reviewer 3

Comment 1: English performances during editing would be necessary.

>Response: We agreed with the suggestion, and we sent the revised manuscript to a native English speaker at American Journal Experts (AJE).  ï¼ˆSee the revised draft marked in green in the revised manuscript)
